# Wharton’s Jelly-Derived Mesenchymal Stem Cells Reduce Fibrosis in a Mouse Model of Duchenne Muscular Dystrophy by Upregulating microRNA 499

**DOI:** 10.3390/biomedicines9091089

**Published:** 2021-08-26

**Authors:** Sang Eon Park, Jang Bin Jeong, Shin Ji Oh, Sun Jeong Kim, Hyeongseop Kim, Alee Choi, Suk-joo Choi, Soo-young Oh, Gyu Ha Ryu, Jeehun Lee, Hong Bae Jeon, Jong Wook Chang

**Affiliations:** 1Stem Cell Institute, ENCell Co., Ltd., Seoul 06072, Korea; earnie.park@encellinc.com (S.E.P.); jb.jeong@encellinc.com (J.B.J.); shin9243@encellinc.com (S.J.O.); sunj-kim98@encellinc.com (S.J.K.); hyeongseop09@encellinc.com (H.K.); ari0621@encellinc.com (A.C.); 2Samsung Medical Center, Stem Cell & Regenerative Medicine Institute, Seoul 06351, Korea; 3Department of Health Sciences and Technology, SAIHST, Sungkyunkwan University, Seoul 06351, Korea; 4Samsung Medical Center, Department of Obstetrics and Gynecology, Seoul 06351, Korea; drmaxmix.choi@samsung.com (S.-j.C.); ohsymd.oh@samsung.com (S.-y.O.); 5Department of Medical Device Management and Research, SAIHST, Sungkyunkwan University School of Medicine, Seoul 06351, Korea; gyuha.ryu@samsung.com; 6The Office of R&D Strategy & Planning, Samsung Medical Center, Seoul 06351, Korea; 7Samsung Medical Center, Department of Pediatrics, Seoul 06351, Korea; jhnr.lee@samsung.com

**Keywords:** Duchenne muscular dystrophy, microRNA-499-5p, skeletal muscle fibrosis, Wharton’s jelly-derived mesenchymal stem cell

## Abstract

The aim of this study was to evaluate the therapeutic effects and mechanisms of Wharton’s jelly-derived mesenchymal stem cells (WJ-MSCs) in an animal model of Duchenne muscular dystrophy (DMD). Mdx mice (3–5 months old) were administered five different doses of WJ-MSCs through their tail veins. A week after injection, grip strength measurements, creatine kinase (CK) assays, immunohistochemistry, and western blots were performed for comparison between healthy mice, mdx control mice, and WJ-MSC-injected mdx mice. WJ-MSCs exerted dose-dependent multisystem therapeutic effects in mdx mice, by decreasing CK, recovering normal behavior, regenerating muscle, and reducing apoptosis and fibrosis in skeletal muscle. We also confirmed that miR-499-5p is significantly downregulated in mdx mice, and that intravenous injection of WJ-MSCs enhanced its expression, leading to anti-fibrotic effects via targeting TGFβR 1 and 3. Thus, WJ-MSCs may represent novel allogeneic “off-the-shelf” cellular products for the treatment of DMD and possibly other muscle disorders.

## 1. Introduction

Duchenne muscular dystrophy (DMD) is a fatal X-linked recessive disorder characterized by progressive loss of muscle mass and function [1]. It occurs in approximately 1/3500 live male births and is caused by mutations in the dystrophin gene that prevent its expression [2]. It is characterized by progressive skeletal and cardiac muscle weakness, with premature death usually occurring at approximately 20 years of age [3,4,5].

There is currently no effective cure for DMD, and patients receive only palliative care [6]. Various therapeutic approaches for DMD have been explored in preclinical and clinical studies. Although clinical trials on DMD are being performed worldwide, to date, no therapies that effectively alter its clinical course have been reported [7]. Furthermore, most gene therapies that are being developed are single-targeted for dystrophin [8,9]. Multitargeted drugs may be more effective since multiple pathogenic mechanisms (e.g., muscle degeneration, inflammation, and fibrosis) are involved in DMD [2,7,10].

Recently, numerous clinical trials have shown that mesenchymal stem cells (MSCs) are safe [11,12] and beneficial treatments for various pathologies [13,14,15]. MSCs are mesoderm tissue-derived cells that can be obtained from autologous or allogeneic sources [16]. These cells can differentiate into adipogenic, chondrogenic, and osteogenic lineages [17,18], and undergo myogenic differentiation upon engraftment in muscle tissues [19,20]. Additionally, various proteins secreted from MSCs have therapeutic effects [21]. These paracrine factors maintain homeostasis by regulating inflammation and immune responses at the sites of lesions, enabling multitargeted therapy [22,23,24]. MSCs from birth-associated tissues, such as human Wharton’s jelly-derived (WJ)-MSCs, may represent very promising stem cell populations for regenerative medicine [25].

Recently, we demonstrated that human WJ-MSCs play an anti-fibrotic role in skeletal muscle fibrosis via matrix metallopeptidase 1 (MMP-1), which acts as a paracrine factor in myotubes [22]. However, it is important to investigate the treatment mechanism in an animal model of DMD. Therefore, in this study, we evaluated the therapeutic effects of WJ-MSCs in an animal model of DMD, and found that WJ-MSCs and derived MMP-1 reduce fibrosis in mdx mice by targeting Transforming Growth Factor-β Receptors (TGFβRs) and enhancing miR-499 expression.

## 2. Materials and Methods

### 2.1. Ethical Statement

This study was approved by the Institutional Animal Care and Use Committee of the Samsung Biomedical Research Institute (SBRI) at the Samsung Medical Center. As an accredited facility of the Association for Assessment and Accreditation of Laboratory Animal Care International, SBRI also abides by the Institute of Laboratory Animal Resources guidelines. In accordance with the guidelines approved by the Institutional Review Board of Samsung Medical Center, umbilical cords were collected with the informed consent of the mothers (IRB #2016-07-102, 20 September 2016).

### 2.2. Animals

The animals used in this study were raised at the Experimental Animal Research Center, which is approved by the Association for Assessment and Accreditation of Laboratory Animal Care. All experiments were performed in accordance with the guidelines of an Animal Research Plan approved by the Institutional Animal Care and Use Committee of Samsung Medical Center. C57BL/10ScSn-Dmdmdx/J (mdx) mice (JAX#001801), and C57BL/10ScSnJ mice (JAX#000476), which act as a control for mdx mice, were purchased from The Jackson Laboratory (Bar Harbor, ME, USA). Mdx mice are spontaneous mutants that do not express dystrophin, a cytoskeletal protein that localizes to the inner face of the sarcolemma, and therefore are a suitable model of DMD. Mdx mouse muscles were histologically normal early in their postnatal development; however, muscle necrosis and visible muscle weakness occurred approximately 1 month later, and the number of differentially expressed genes in the mdx mice peaked at 8–12 weeks old, indicating that this is the period of maximum muscle regeneration. Therefore, all experimental procedures were performed in male and female mice that were at least 3 months old.

### 2.3. WJ-MSC and MMP-1 Administration

WJ-MSCs were isolated as previously described [23] and cultivated according to the standard operating procedures of the good manufacture practice facility at Samsung Medical Center. Mdx mice (3–5 months old) were injected with 5 × 10^3^ to 5 × 10^5^ WJ-MSCs (suspended in 100 µL of phosphate-buffered saline (PBS)) through the lateral tail vein. Because clumping cells can cause embolisms in the blood vessels, they were separated by mixing prior to injection and injected as slowly as possible. Human recombinant MMP-1 (200 ng) was injected into the gastrocnemius muscles. The mice were sacrificed with isoprene 7 d after administration, and their gastrocnemius muscles were used in further experiments.

### 2.4. Grip Strength Measurements

The grip strength experiment measures the muscle strength of a mouse using a grip strength meter (BIO GS3, BIOSEB, Vitrolles, France). It is an in vivo assay commonly used to measure disease progression and decreases in fore- and/or hindlimb muscle strength in mdx mice.

### 2.5. CK Assays

Blood collected by retro-orbital plexus bleeding was incubated at 22–24 °C for >1 h and centrifuged at 15,500× *g* for 10 min to collect the upper layer of clear serum, which was used in the experiment. CK activity was determined using the Creatine Kinase Activity Assay Kit (Colorimetric) (ab155901, Abcam, Cambridge, UK).

### 2.6. IHC, Sirius Red Staining, and H&E Staining

After the euthanized mice were skinned, the gastrocnemius muscles and diaphragms were carefully removed for fixation in 4% paraformaldehyde for 24 h. Then, the muscles were embedded in paraffin, sectioned to 4 μm, and stained with Sirius Red to observe muscle fibrosis and hematoxylin and eosin to observe muscle histopathology and central nuclei as features of mdx mouse muscle. Muscle restoration was detected by incubating fixed gastrocnemius sections with myosin heavy chain antibody (#MAB4470, at 1/1000 dilution; R&D Systems, MN, USA), annexin V antibody (ab14196, at 1/500 dilution, Abcam), and fibronectin antibody (ab2413, at 1/200 dilution, Abcam) for 18 h at 4 °C, then with Alexa Fluor^®^ 488 AffiniPure Goat Anti-mouse IgG (H+L) (A10680, Thermo Fisher Scientific, Waltham, MA, USA), Alexa Fluor^®^ 488 AffiniPure Goat Anti-rabbit IgG (H+L) (A27034, Thermo Fisher Scientific), and Alexa Fluor^®^ 594 AffiniPure Goat Anti-rabbit IgG (H+L) (A11037, Thermo Fisher Scientific) secondary antibodies. Sections were counterstained with Hoechst 33342 (H1339, Thermo Fisher Scientific).

### 2.7. PCR

PCR was performed in 20 μL reactions containing 10 μL of 2× Power SYBR™ Green PCR Master Mix (4367659, Thermo Fisher Scientific), 1 μL of 10 pmol/μL each primer, and 8 μL DNA at various concentrations. PCR was performed using a QuantStudio 6 Flex Real-Time PCR System (Thermo Fisher Scientific) in MicroAmp Optical 384-Well Reaction Plates (4326270, Thermo Fisher Scientific). The program included an initial denaturation step at 95 °C for 10 min, followed by 40 cycles of 95 °C for 15 s and 68 °C for 30 s. A melting curve consisting of 15 s at 95 °C, 1 min at 60 °C, and 15 s at 95 °C was obtained after each assay to verify the specificity of the reaction.

The Alu PCR primers 5′-GTCAGGAGATCGAGACCATCCC-3′ (forward (F)) and 5′-TCCTGCCTCAGCCTCCCAAG-3′ (reverse (R)) were designed based on the sequence of plasmid pPD39 (Ya5 subfamily) using the program Oligos© 1999–2002, designed by R. Kalendar of the Institute of Biotechnology, University of Helsinki. Primers were purchased from Bioneer (Bioneer, Daejoen, Korea).

### 2.8. miRNA Sequencing and Data Analysis

Total RNA was extracted from the skeletal muscles of healthy and mdx mice using TRIzol Reagent (Invitrogen, Waltham, MA, USA). RNA quality was assessed using an Agilent 2100 Bioanalyzer and RNA 6000 Pico Chips (Agilent Technologies, Amsterdam, The Netherlands), and RNA concentrations were measured on a NanoDrop 2000 Spectrophotometer (Thermo Fisher Scientific).

Library construction was performed using the NEBNext Multiplex Small RNA Library Prep kit (New England BioLabs, Inc., Ipswich, MA, USA), according to the manufacturer’s instructions. High-throughput sequences were produced on the NextSeq500 system (Illumina, San Diego, CA, USA) using single-end sequencing of 75 bp reads.

Sequence reads were mapped using Bowtie 2, and mature miRNA sequences were used as a reference for mapping. Read counts mapped on mature miRNA sequences were extracted from the alignment file using bedtools (v2.25.0) and Bioconductor, which uses the R statistical programming language (version 3.2.2; R development Core Team, 2011). The quantile normalization method was used for comparisons between the samples. MeV (version 4.9.0) was used for heatmap clustering.

### 2.9. miRNA Isolation and qPCR

miRNAs were isolated from skeletal muscle tissues or cells using the mirVana™ miRNA Isolation Kit (Invitrogen), according to the manufacturer’s instructions. After isolation, miRNA concentrations were measured using the Qubit microRNA Assay kit and a Qubit 4 fluorometer (Invitrogen). The isolated miRNAs were reverse transcribed into cDNA at a concentration of 5 ng/μL using the TaqMan™ Advanced miRNA cDNA Synthesis Kit (Applied Biosystems, Waltham, MA, USA).

We performed qPCR on a QuantStudio™ 6 Flex Real-Time PCR System (Applied Biosystems) using TaqMan™ Fast Advanced Master Mix (Applied Biosystems) in fast-cycling mode. Taqman™ Advanced miRNA assays (Invitrogen) were used to measure the levels of hsa-miR-499-5p (Assay ID: 478139_mir) and hsa-miR-26a-5p (Assay ID: 477995_mir). All reactions were performed in triplicate, and comparative quantification of each target gene was performed based on the cycle threshold (CT), which was normalized to miR-26a-5p using the ΔΔCT method proposed by Livak and Schmittgen.

### 2.10. RNA Isolation and qPCR

Total RNA was isolated from skeletal muscle tissues using TRIzol Reagent, following the manufacturer’s instructions. SuperScript IV Reverse Transcriptase (Invitrogen) was used to convert RNA to cDNA. The mRNA expression levels of the target genes were measured using 2× Power SYBR Green Master Mix (Applied Biosystems), with an initial denaturation at 95 °C for 10 min followed by 40 cycles of 15 s at 95 °C and 60 s at 55 °C. The transcript levels of each gene were normalized to GAPDH. The following primers were used: mouse GAPDH F: 5′-CATGGCCTTCCGTGTTCCTA-3′, R: 5′-CCTGCTTCACCACCTTCTTGAT-3′, mouse TGFβR1 F: 5′-ATGGGCTTAGTGTTCTGG-3′, R: 5′-CCTGTTGGCTGAGTTGTG-3′, mouse TGFβR3 F: 5′-GGAGGTGCATGTCCTGAATC-3′, and R: 5′-CAGACTTGTGGTGGATGTGG-3′.

### 2.11. Western Blot Analysis

For total protein extraction, myotubes were scraped from the culture dishes and lysed in ice-cold radioimmunoprecipitation buffer (9.8 mol/L UREA, 4% CHAPS, 130 mmol/L dithiothreitol, 40 mmol/L Tris-HCl, 0.1% sodium dodecyl sulfate, 1 mmol/L EDT, and a protease/phosphatase inhibitor cocktail). Protein quantification was performed using the Bradford assay (Bio-Rad Laboratories, Hercules, CA, USA). Equivalent amounts of protein were resolved by sodium dodecyl sulfate-polyacrylamide gel electrophoresis and transferred onto polyvinylidene difluoride membranes (Bio-Rad Laboratories). After blocking with 5% skim milk, membranes were probed with primary antibodies against fibronectin (ab2413, at 1/10,000 dilution, Abcam), myosin heavy chain (#MAB4470, 1/10,000 dilution; R&D Systems), p-smad2/3 (8828S, 1/1000 dilution, Cell Signaling Technology, Danvers, MA, USA), smad2/3 (3102S, 1/1000 dilution, Cell Signaling Technology), GAPDH (sc-32233, 1/5000 dilution; Santa Cruz Biotechnology, Dallas, TA, USA) and β-actin (sc-47778, 1/10,000 dilution; Santa Cruz Biotechnology) overnight at 4 °C. The membranes were washed three times with TBST and incubated with secondary antibodies: a goat anti-rabbit IgG HRP-conjugated antibody (GTX213110-01, 1/10,000 dilution; GeneTex, Irvine, CA, USA) or a goat anti-mouse IgG HRP-conjugated antibody (GTX213111-01, 1/10,000 dilution; GeneTex) for 1 h at room temperature. Band intensities were quantified using Image Lab software (Bio-Rad Laboratories).

### 2.12. Cell Culture

The mouse myoblast cell line C2C12 (ATCC CRL-1772, American Type Culture Collection, Manassas, VA, USA) was cultured in Dulbecco’s modified Eagle’s medium (Biowest S.A.S, Nuaille, France) supplemented with 10% fetal bovine serum (Gibco BRL, MA, USA), 100 U/mL penicillin, and 100 μg/mL streptomycin (Gibco BRL) in 5% CO_2_ at 37 °C. For myotube differentiation, the culture medium was replaced with differentiation medium supplemented with 5% horse serum (Gibco BRL) for 5 d. After differentiation, 2 mM H_2_O_2_ (Sigma, MO, USA) was added to the growth medium for 24 h to induce fibrosis.

In TGFβ receptor inhibition experiments, cells were treated with 10 µM galunisertib (LY2157299, S2230, Selleck Chemicals, Houston, TX, USA) for 24 h in serum-free medium.

### 2.13. miRNA Mimic Transfection

To determine the effects of miR-499-5p in the myotube fibrosis model, myotubes were transfected with 30 nmol/L human miR-499-5p mimic (GenePharma, Shanghai, China) using HiPerFect Transfection Reagent (Qiagen, Inc., Hilden, Germany) for 24 h in serum-free medium.

### 2.14. Immunocytochemistry

Myotubes were fixed with 4% paraformaldehyde. Picro-Sirius Red staining (Abcam) was performed to detect collagen deposition in each group, and the images were acquired by light microscopy. For immunocytochemistry, myotubes were incubated with PBST (PBS containing 0.25% Triton X-100) to increase permeability and blocked with blocking solution (PBST with 4% normal goat serum and 0.1% bovine serum albumin (BSA)), then incubated with anti-collagen I antibody (Abcam) overnight at 4 °C. After three washes with PBST, myotubes were incubated with secondary antibody in 0.1% BSA in PBS. Hoechst was used to stain nuclei. Images were acquired on a Zeiss LSM 700 confocal microscope (Carl Zeiss, Oberkochen, Germany), and quantified using ImageJ (version 1.53e, US National Institutes of Health, Bethesda, MD, USA).

### 2.15. Statistical Analyses

All results were analyzed by one-way analysis of variance or independent t-tests using SPSS software (version 23.0; IBM, Armonk, NY, USA). *p* < 0.05 was considered statistically significant, and data are expressed as the mean ± the standard error of the mean.

## 3. Results

### 3.1. Therapeutic Effects of WJ-MSCs in Mdx Mice

The therapeutic effects of WJ-MSC injection were confirmed using grip strength measurements, creatine kinase (CK) assays, Sirius Red assays, and immunohistochemistry (IHC). WJ-MSCs (5 × 10^3^ (dose 1), 1 × 10^4^ (dose 2), 5 × 10^4^ (dose 3), 1 × 10^5^ (dose 4), and 5 × 10^5^ (dose 5)) were administered through the tail vein. A week after injection, grip strength was compared between healthy mice, control mdx mice, and WJ-MSC-injected mdx mice. We observed increased strength in the injection group compared to the control mdx group after doses 1–5; however, doses 4 and 5 induced minimal or no increase in strength compared to dose 3 (Figure 1a,b). The symptom-relieving effects of WJ-MSCs were examined by measuring the level of serum CK, an indicator of muscle disease. CK levels were reduced in the cell injection group compared with the untreated mdx group at all doses, with significant increases after doses 3–5 compared to after lower doses (Figure 1c). The degree of fibrosis in mdx mouse muscle was determined by Sirius Red staining, and fibrosis mitigation was confirmed in the dose 3 group (Figure 1d,e). The muscle-regenerating and anti-apoptotic effects of WJ-MSCs were observed through myosin heavy chain and annexin V staining, respectively, with significant effects after dose 3 and similar or slightly increased effects at higher doses (Figure 1d,f,g). From these results, we determined that dose 3 was optimal, as it was the smallest dose resulting in treatment effects, and fibrosis relief, muscle regeneration, and anti-apoptotic effects did not increase significantly at higher doses.

### 3.2. MicroRNA (miRNA) Sequencing of Mdx Mouse Skeletal Muscle Identifies Downregulation of miR-499-5p

Numerous studies have reported pathological symptoms in mdx mice, and there are many differences between them and healthy mice. To identify differences in miRNA expression between healthy and mdx mice, miRNA sequencing was performed on gastrocnemius muscles from each group. We found several downregulated miRNAs in mdx mice compared to healthy mice (Figure 2a). We focused on miR-499-5p, which was the most downregulated miRNA in mdx mice.

To confirm the miRNA sequencing results, we assessed the miR-499-5p expression levels in each group using quantitative polymerase chain reaction (qPCR), again using gastrocnemius muscles from each group. The expression level of miR-499-5p in mdx mice was significantly lower than in healthy controls. However, it was significantly increased after WJ-MSC administration (Figure 2b).

Taken together, these results confirm that miR-499-5p is an important downregulated miRNA in mdx mice and that intravenous (IV) injection of WJ-MSCs enhances its level. In our previous study, we found that MMP-1 of human WJ-MSCs alleviated skeletal muscle fibrosis in mdx mice and in an in vitro fibrosis model [22]. Therefore, we administrated human MMP-1 protein (200 ng) through intramuscular (IM) injection in the gastrocnemius muscles of mdx mice and observed that MMP-1 enhances miR-499-5p expression level (Appendix A).

Therefore, to determine whether the increase in miR-499-5p caused by WJ-MSC injection is the main mechanism regulating their positive effects in the DMD model, we examined the effects of miR-499-5p in an in vitro fibrosis model.

### 3.3. miR-499-5p Directly Targets Transforming Growth Factor β Receptor (TGFβR)1 and TGFβR3

TargetScan detected highly conserved miR-499-5p targeted sites in mouse TGFβR1 and TGFβR3. As TGFβ is a central mediator of fibrosis that acts through TGFβRs, we examined TGFβR1 and TGFβR3 transcript levels in control and mdx mice using qPCR of gastrocnemius muscles from each group. The transcript levels of TGFβR1 and TGFβR3 were significantly increased in mdx mice compared to healthy controls, and significantly decreased after WJ-MSC administration (Figure 3). Collectively, these results demonstrate that miR-499-5p targets TGFβR1 and TGFβR3, which are key factors in fibrosis.

### 3.4. Anti-Fibrotic Effect of WJ-MSCs in the Skeletal Muscle of Mdx Mice

We next investigated the anti-fibrotic effects of WJ-MSCs. We injected mice with 5 × 10^4^ cells through the tail vein and performed Sirius Red assays and fibronectin IHC. Sirius Red staining revealed increased fibrosis in mdx mice compared to control mice, which was decreased in WJ-MSC-injected mdx mice. Fibronectin was increased in the control mdx mice compared to the healthy mice and was decreased in the injection group. (Figure 4a–c). Additionally, the Smad2/3 phosphorylation pattern was consistent with fibronectin expression (Figure 4d,e).

To analyze the distribution of WJ-MSCs in mouse tissues, we quantified the WJ-MSCs present in the gastrocnemius muscle, heart, liver, lung, and spleen 1 week after IV injection by qPCR. When equal amounts of cells were injected into control and mdx mice, mdx mice retained more cells in their skeletal muscle than control mice (Appendix A). These results suggest that IV-administered WJ-MSCs specifically migrate to the muscles of mdx mice.

Also, to determine the duration of effect by MSC injection, we evaluated the degree of anti-fibrotic effect over time. Fibrosis was observed through Sirius red staining, and the results of MSC injection showed an anti-fibrotic effect consistently from 4 days to 4 weeks later (Appendix A).

### 3.5. Anti-Fibrotic Effect of WJ-MSCs in the Diaphragm of Mdx Mice

Additionally, the anti-fibrotic effects of MSC were observed in the diaphragm. We injected mice with 5 × 10^4^ cells through the tail vein and performed H&E and Sirius Red staining and fibronectin IHC. H&E and Sirius red staining confirmed that the fibrosis area, which had been increased in the mdx mice compared to normal mice, decreased after MSC injection (Figure 5a–c).

### 3.6. Anti-Fibrotic Effect of miR-499-5p in H_2_O_2_-Treated Myotubes

To investigate the role of miR-499-5p in the fibrosis observed in mdx mice, we used a previously described in vitro myotube fibrosis model [22]. Briefly, differentiated myotubes were treated with 2 mM H_2_O_2_ to induce a fibrotic phenotype (Figure 6e, Appendix A). This concentration of H_2_O_2_ used did not induce cell death in the myotubes (data not shown). To determine the effect of miR-499-5p, myotubes were transfected with an miR-499-5p mimic (30 nmol/L) for 24 h (Appendix A). The miR-499-5p mimic attenuated damage to the myotubes.

Relative fibrosis intensity was measured in two ways. Sirius Red staining was used to detect collagen accumulation. The relative collagen intensity of myotubes was significantly increased in the H_2_O_2_-treated group compared to that in the control group (Figure 6a,c). In addition, IHC was used to detect collagen I deposition in the myotubes, and this was also significantly increased in the mimic-treated group. However, transfection of the miR-499-5p mimic attenuated myotube fibrosis, with significantly decreased collagen deposition in the miR-499-5p group compared to the non-transfected group (Figure 6b,d). These data demonstrate that H_2_O_2_ treatment induces fibrosis in myotubes, and that an miR-499-5p mimic can ameliorate this degeneration. Consistently, Western blotting analysis showed significantly decreased fibronectin and increased MHC levels in the miR-499-5p group compared to the H_2_O_2_-treated group. For confirming that miR-499-5p directly targets the TGFβ receptor, we treated fibrosis-induced myotubes with the TGFβ receptor inhibitor galunisertib. Similar to the results of the miR-499-5p treatment, the protein levels of decreased fibronectin and increased MHC levels were observed in the galunisertib group. Additionally, the increased phospho-smad2/3 levels in fibrosis myotube were decreased in the miR-499-5p and galunisertib groups (Figure 6f,g). These results indicate that miR-499-5p produces an anti-fibrotic effect by inhibiting the TGFβ signaling pathway.

## 4. Discussion

In this study, we have demonstrated that a single IV administration of WJ-MSCs to 3–5-month-old mdx mice has therapeutic effects in skeletal muscle, such as CK reduction, behavior recovery, muscle regeneration, apoptotic evasion, and reduced fibrosis in skeletal muscle and diaphragm. To the best of our knowledge, this work is novel from three standpoints: it is the first to demonstrate dose-dependent effects of WJ-MSCs in an mdx mouse model, show treatment effects via a single IV administration rather than local administration, and evaluate changes in miRNA expression in the skeletal muscles of mdx mice.

We aimed to determine the optimal MSC dose in the mdx mouse model, to ease their translation into clinical settings. Higher-than-optimal dosing would not increase the treatment efficacy. Therefore, it would be logical to establish an optimal dose-response relationship to select the optimal dose from a cost-effective point of view. To this end, five different doses of WJ-MSCs were compared for effects on the CK level, behavioral tests, and histological assessments. Higher doses have resulted in better therapeutic effect only until the 3rd highest dose (dose 3) among the 5 doses tested by us. Therefore, dose 3 will be considered for the clinical translation in further studies.

We studied the distribution of IV-injected WJ-MSCs and how they acted in the skeletal muscle. Similar to the biodistribution evaluation in our previous study, IV-injected WJ-MSCs mainly accumulated in the livers of healthy mice and not in the skeletal muscle. However, injected WJ-MSCs were retained in the skeletal muscles of mdx mice. As reported in many studies, MSCs migrate to pathogenic sites in response to inflammation, resulting in therapeutic effects [26,27,28]. Therefore, for the treatment of DMD, which involves muscle inflammation, IV administration seems to be a possible route that, allows the systemic transfer of MSCs to the skeletal muscle. Despite the therapeutic effect shown in mdx mice, the overall number of cells remaining in mdx mice is lower than in normal mice; this effect can be attributed to MSC damage upon exposure to the DMD environment. A follow-up study is to be conducted for maximizing the treatment efficacy through a strategy for increasing the survival of MSCs exposed to the DMD environment.

It was recently reported that miRNAs are expressed in specific body compartments [29,30]. Several miRNAs are specifically expressed in muscle cells, where they play crucial roles in proper muscle development and function [31,32]. Moreover, altered miRNA levels have been observed in several muscular disorders, such as DMD and other myopathies [31,33,34,35]. In this study, we showed that miR-499 expression was decreased in the skeletal muscle of mdx mice compared to control mice and recovered in MSC-administered mice. In addition, we confirmed that MMP-1, which was reported in the previous study as being related to reducing skeletal muscle fibrosis [22], increased the level of miR-499 in skeletal muscle. Additionally, we found that miR-499 reduced skeletal muscle fibrosis by down-regulating TGFβRs through inhibition of SMAD2 and 3 phosphorylation.

Patients with DMD display persistent skeletal muscle fibrosis with characteristic muscle inflammation, apoptosis, and repair, which play significant roles in disease progression and reduce life expectancy [36]. Inflammation is thought to precede TGFβ overexpression and actual muscle wasting [37,38]. In particular, the progressive muscle wasting, weakness, and loss of muscle mass observed in patients suffering from these diseases were shown to be accompanied by increased mRNA [39] and protein levels of TGFβ [40] in DMD. Thus, new advances in targeting TGFβ, targeting fibrosis may prove an important strategy for DMD treatment, as there are already many approved anti-fibrotic drugs on the market that target and modulate fibrotic signaling [36,41,42]. Targeting profibrotic growth factors or cytokines to slow fibrosis development has also shown promising results. Members of the TGFβ family elicit cellular responses via TGFβRs, to activate the canonical SMAD pathway [43,44]. Improved muscle repair can be achieved not only by reducing fibrosis development but also by modifying the inflammatory response [45,46]. Upregulation of TGFβ activity accompanies inflammation and fibrosis and abolishes the homeostasis required for proper and efficient muscle regeneration [42,47]. Thus, inhibition of TGFβ by targeting TGFβRs could modify the proliferation of myoblasts, fibroblasts, and inflammatory cells found within injured muscle [46,47,48]. Similarly, we observed that the enhancement of mir-499 expression reduced skeletal muscle fibrosis, as well as improved muscle regeneration through increased levels of MHC (Figure 1 and Figure 6). Collectively, our findings suggest that human WJ-MSCs and derived MMP-1 play an anti-fibrotic role in skeletal muscle fibrosis by targeting TGFβRs with enhancing miR-499 expression (Figure 7).

## 5. Conclusions

Our group has reported that human WJ-MSCs exert anti-apoptotic, inflammatory, and immunomodulatory effects on the surrounding cells through the secretion of paracrine factors; WJ-MSCs have been suggested as an effective therapeutic agent for muscle diseases. In these studies, our data suggest that single IV injections of WJ-MSCs have multisystem therapeutic effects in a mouse model of DMD, including CK reduction, behavior recovery, muscle regeneration, and decreased apoptosis and fibrosis. In addition, we confirmed that human WJ-MSCs reduce fibrosis in DMD by up-regulating miR-499.

Therefore, WJ-MSCs represent novel allogeneic “off-the-shelf” cellular products that could be used to treat DMD and possibly other muscle disorders.

## Figures and Tables

**Figure 1 biomedicines-09-01089-f001:**
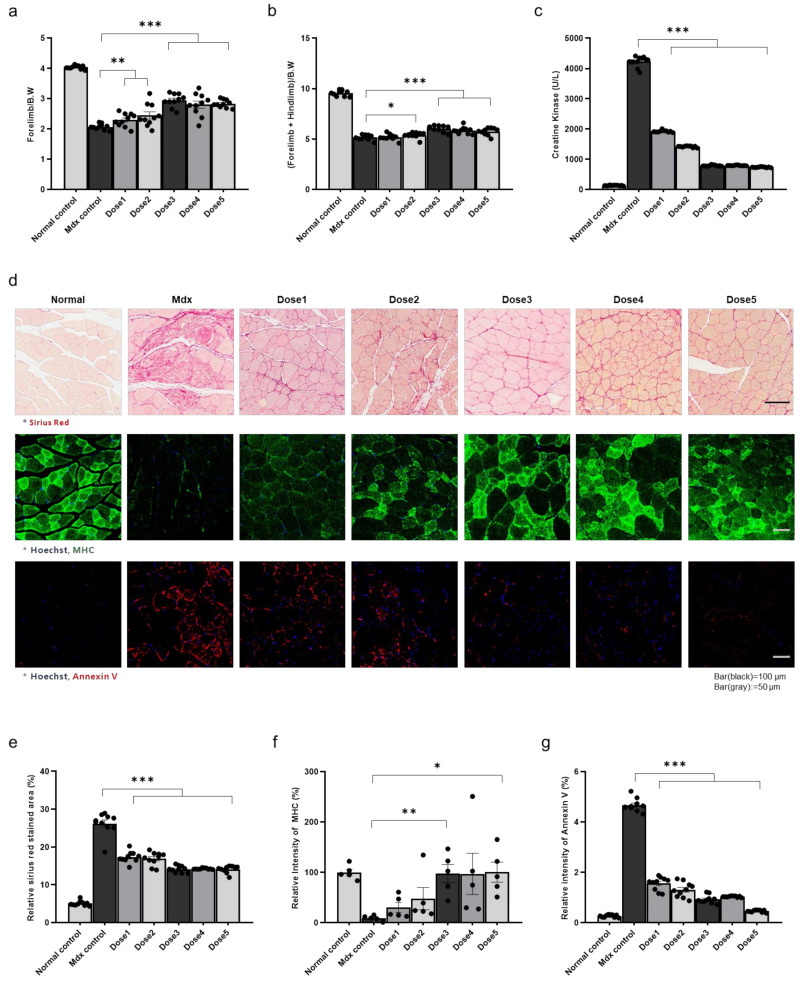
Therapeutic effects of Wharton’s jelly-derived mesenchymal stem cells (WJ-MSCs) in the skeletal muscle of mdx mice (**a**) Forelimb grip strength, (**b**) forelimb + hindlimb grip strength, (**c**) creatine kinase (CK) levels, (**d**) fibrotic areas, and myosin heavy chain and annexin V expression levels were measured in control, untreated mdx mice, and WJ-MSC-treated mdx mice. (**e**–**g**) Quantitative measurements were performed in ImageJ. The data are expressed as the mean ± SEM. Bars with different superscripts indicate significantly different values (* *p* < 0.05, ** *p* < 0.01, *** *p* < 0.001).

**Figure 2 biomedicines-09-01089-f002:**
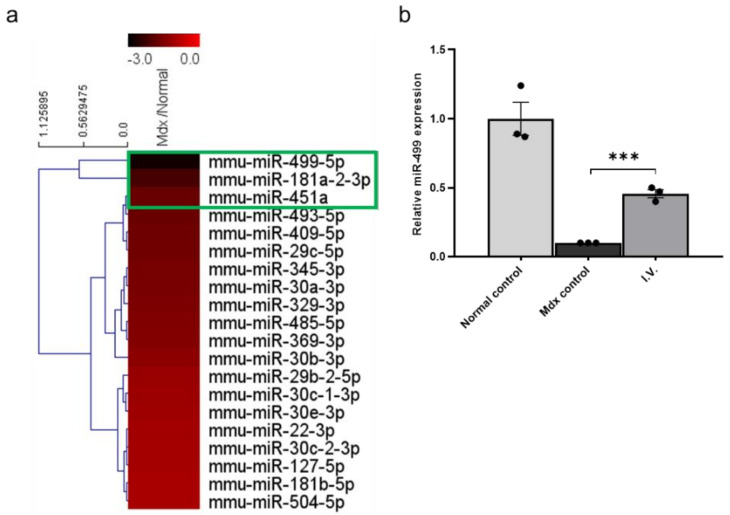
miR-499-5p is downregulated in mdx mice compared to healthy mice (**a**) Heatmap depicting the levels of miRNAs commonly expressed in humans and mice in mdx mice and healthy mice. (**b**) Relative miR-499-5p expression levels in the skeletal muscles of each group. miR-26a-5p was used as an endogenous control. The data are expressed as the mean ± SEM. Bars with different superscripts indicate significantly different (*** *p* < 0.001).

**Figure 3 biomedicines-09-01089-f003:**
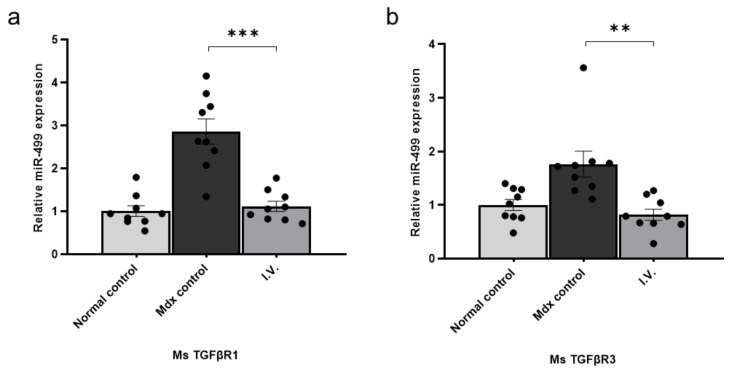
Transforming growth factor β receptor (TGFβR)1 and TGFβR3 are direct targets of miR-499-5p (**a**) Relative TGFβR1 expression levels were measured in skeletal muscles from each group. Glyceraldehyde 3-phosphate dehydrogenase (GAPDH) was used as an endogenous control, and data are expressed as the mean ± the SEM. Bars with different superscripts indicate significantly different values. (**b**) Relative TGFβR3 expression levels were measured in the skeletal muscle of each group. GAPDH was used as an endogenous control, and data are expressed as mean ± SEM (** *p* < 0.01, *** *p* < 0.001).

**Figure 4 biomedicines-09-01089-f004:**
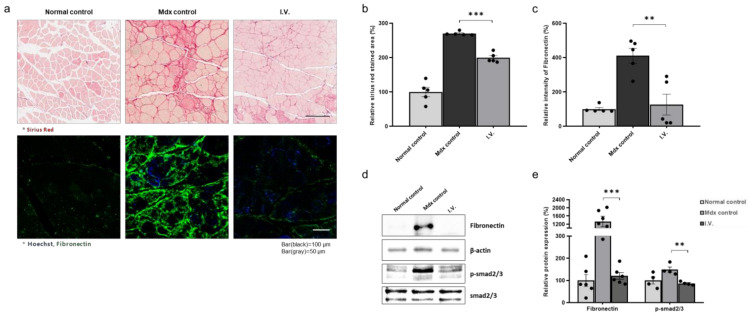
Anti-fibrotic effect of Wharton’s jelly-derived mesenchymal stem cells in the skeletal muscle of mdx mice (**a**) Representative images of Sirius Red staining for fibrosis evaluation and immunohistochemistry for fibronectin detection. (**b**,**c**) Quantitative measurements were performed in ImageJ, and the data are expressed as the mean ± SEM. Bars with different superscripts indicate significantly different values. (**d**) Representative western blot for fibronectin and p-smad2/3. β-actin was used as a loading control. Phospho-smad2/3 was normalized to total smad2/3. (**e**) Quantification of fibronectin and phospho-smad2/3 protein levels. The data are expressed as the mean ± SEM. Bars with different superscripts indicate significantly different values (** *p* < 0.01, *** *p* < 0.001).

**Figure 5 biomedicines-09-01089-f005:**
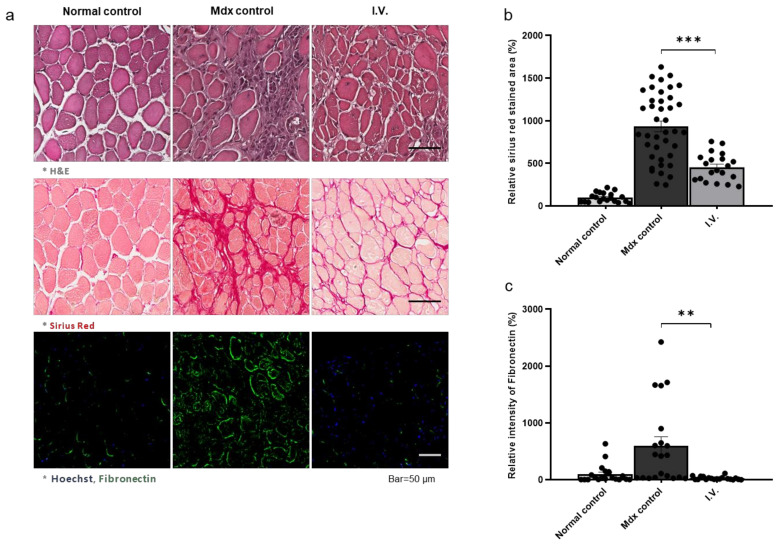
Antifibrotic effect of Wharton’s jelly-derived mesenchymal stem cells in the diaphragm of mdx mice (**a**) Representative images of H&E and Sirius red staining and immunohistochemistry for fibronectin detection. (**b**,**c**) Quantitative measurements were performed in ImageJ, and the data are expressed as the mean ± SEM. Bars with different superscripts indicate significantly different values (** *p* < 0.01, *** *p* < 0.001).

**Figure 6 biomedicines-09-01089-f006:**
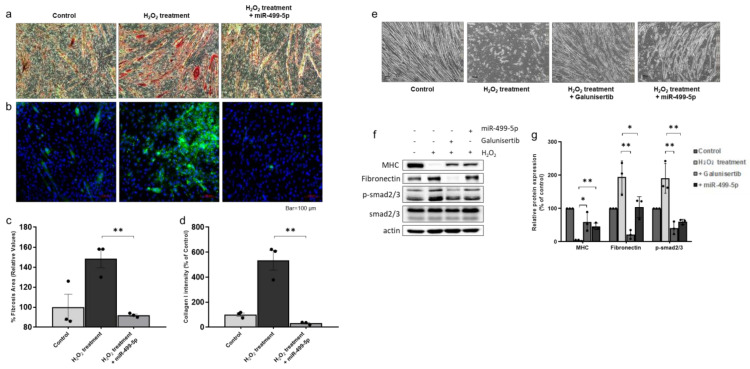
Effects of miR-499-5p on an H_2_O_2_-induced myotube fibrosis model (**a**,**c**) Fibrosis was assessed as the percentage of the area positive for Sirius Red staining. Quantitative measurements were performed in ImageJ, and data are expressed as the mean ± SEM. Bars with different superscripts indicate significantly different values (** *p* < 0.01). Scale bars: 100 μm. (**b**,**d**) Immunocytochemistry to detect collagen I deposition in myotubes. Quantitative measurements were performed in ImageJ, and data are expressed as mean ± SEM. Bars with different superscripts indicate significantly different values (** *p* < 0.01). Scale bars: 100 μm. (**e**) Differentiated myotubes were treated with 2 mM H_2_O_2_ in the absence or presence of galunisertib and miR-499-5p mimic. Administration of galunisertib and miR-499-5p mimic induced recovery of myotubes damaged by H_2_O_2_ treatment (bar = 100 µm). (**f**) Representative western blots for myosin heavy chain, fibronectin, and phospho-smad2/3. β-actin was used as a loading control. Phospho-smad2/3 was normalized to total smad2/3. (**g**) Quantification of myosin heavy chain, fibronectin and phospho-smad2/3 protein levels. Data are expressed as the mean ± SEM. Bars with different superscripts indicate significantly different values (* *p* < 0.05, ** *p* < 0.01).

**Figure 7 biomedicines-09-01089-f007:**
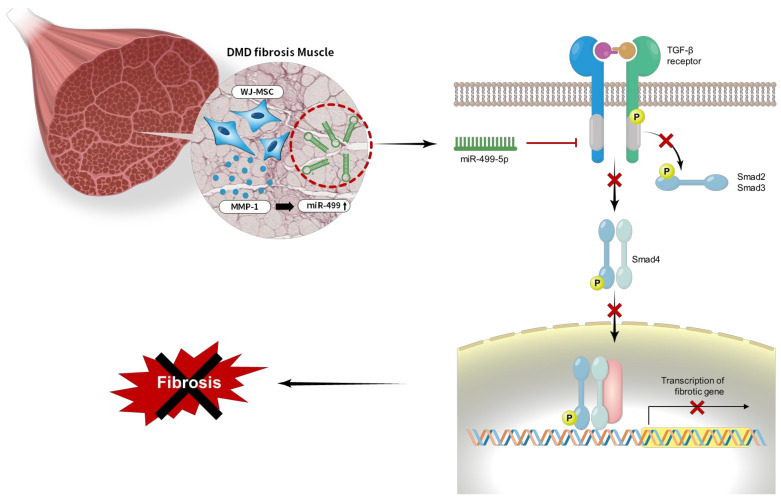
A model of miR-499-dependent anti-fibrotic signaling pathway. Overall proposed model of the anti-fibrotic mechanism of miR-499-5p in skeletal muscles in Duchenne muscular dystrophy (DMD). The activation of TGF β signaling pathway induces fibrosis. Our findings suggest that MMP-1 secreted from WJ-MSCs increases miR-499 expression level. Increased miR-499-5p down-regulates to the TGFβ receptor I and blocks smad2/3 phosphorylation, leading to fibrotic muscle recovery by the inactivated TGFβ signaling pathway (
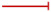
: block, 
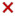
: inactivation).

## Data Availability

The data that support the findings of this study are available from the corresponding author upon reasonable request.

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
