# Peer review of "Wharton’s Jelly-Derived Mesenchymal Stem Cells Reduce Fibrosis in a Mouse Model of Duchenne Muscular Dystrophy by Upregulating microRNA 499"

_biomedicines, 2021, doi:10.3390/biomedicines9091089_

Round 1

Reviewer 1 Report

Authors made corrections and inserted some modifications according my critical points. They put a lot of effort in to improve the manuscript.

The figures are improved and contain statistical evaluations. Inclusion of Figure 7. helps to understand the mechanism by which WJ-MSCs and derived MMP-1 might reduce fibrosis by targeting TGFβRs and enhancing miR-499 expression.

These modifications and additional data improved the manuscript extensively.

Reviewer 2 Report

Authors performed additional experiments and revised their manuscripts according to the reviewers's comments. This reviewer has no critical comments on the revised manuscript.

This manuscript is a resubmission of an earlier submission. The following is a list of the peer review reports and author responses from that submission.

Round 1

Reviewer 1 Report

Authors previously reported that intravenously injected Wharton's jelly-derived mesenchymal stem cells (WJ-MSCs) reduced muscle fibrosis in dystrophin-deficient mdx mice, a widely used DMD model mice, by secretion of MMP-1 (Choi et al., International Journal of Molecular Sciences 2020, 21,6269: ref. 22). In the present study, authors report that miR-499-5p is down-regulated in mdx muscle. IV injection of WJ-MSCs restored miR-499-5p expression in mdx mice. TargetScan software has suggested that miRNA-499-5p targets TGFβR1 and TGFβ3. Therefore, authors propose that WJ-MSCs reduced fibrosis in mdx muscle via miR-499-5p-mediated blockage of TGF-β signaling. Findings are potentially interesting, but this study is preliminary to prove their hypothesis.

Comments:

  1. Figure 1. Sirius red staining is not clear. H.E. staining is also informative. Myosin heavy chain (MHC) signal is at the periphery of myofibers, suggesting non-specific staining. Which type of MHCs (embryonic, neonatal or adult type) was stained? Which cell type was Annexin V-positive? Myonuclei are positive for Annexin V?
  2. Authors examined mdx muscle after 1 week of treatment. It is important to evaluate the longer-term therapeutic effects of WJ-MSCs.
  3. Authors examined only the gastrocnemius muscle for fibrosis, which regenerates well and shows minimal fibrotic changes at 3-6 months of age. The diaphragm is the best muscle to examine, because it’s the most affected muscle with extensive fibrosis.
  4. Figure 3. The data show that TGFβR1 and TGFβR3 were upregulated in mdx mice and their expression was reduced by i.v. of WJ-MSCs. Did authors administrate miR-499-5p? Are Smads less phosphorylated in mdx skeletal muscle of iv group than non-treated mdx muscle? Based on the present data, is it possible to conclude that TGFβR1 and TGFβR3? are direct targets of miR-499-5p?
  5. Figure 5. Did miR-499-5p reduce the levels of TGFβR1 and TGFβR3? Did authors examine the phosphorylation levels of Smads? Do TGF-β inhibitors have the same effects on myotubes as miR-499-5p transfection?
  6. In vivo, mesenchymal cells are mainly responsible for fibrosis. Does their in vitro fibrosis model using C2C12 myotubes reflect in vivo pathology?
  7. Are the expression and functions of MMP1 and of miR-499-5p related? Do authors have data suggesting a relationship between MMP-1 and miR-499-5p?
  8.  

Minor comments:

  1. Figure 2. Authors used miR-26a-5p to normalize the expression level of miR-499-5p. The expression level of miR-26a-5p was stable after I.V. treatment?
  2. Figure legends lack information to understand the experiments.
  3. Supplementary Table 1. Authors detected human genomic DNA in several organs of mdx mice by Alu PCR. Please indicate the denominator. 2.73 ng/XX. How many mice were examined?
  4. Supplementary Figure 1. It is informative to show the miR-499-5p level of the control.

Reviewer 2 Report

Review of the paper „Wharton's jelly-derived mesenchymal stem cells reduce fibrosis in a mouse model of Duchenne muscular dystrophy by up-regulating microRNA 499”

Sang Eon Park et al, Biomedicines 2021

This article evaluates the possible therapeutic effects and mechanisms of Wharton's jelly-derived mesenchymal stem cells (WJ-MSCs) in an animal model of Duchenne muscular dystrophy, the mdx mice.

During the past years, mesenchymal stem cells (MSCs) were considered as potential actors in various regenerative medicine applications.

Umbilical cord has proved to be a unique source of MSCs, which are abundantly present in its tissue/matrix, the Wharton’s jelly. According to several publications, these cells can differentiate toward ectoderm-, mesoderm-, and endoderm- derived cell types. Due to their high differentiation and low immunogenecity potential and easy access they are recently considered as an excellent source for advanced therapy of various diseases.

In highlight of these advantages authors focused on the possible therapy of the devastating Duchenne Muscular Dystrophy (DMD) disorder by treating the original animal model, the mdx mouse with WJ-MSCs.

Special remarks:

Lines 42-43: “Targeting only dystrophin” is not appropriate as this approach should be the key element. All other pathogenic mechanisms (as muscle degeneration, inflammation, and fibrosis) are straight-forward consequences of the dystrophin deficiency and as such, improvement of these factors would be expected as well.

Line 69: “doses 4 and 5 induced minimal or no increase in strength” – needs inclusion of “further increase” as there is definitely an increase which is indeed not higher than with doses 2 and 3.

Figures 1-4. do not mention in the legend which statistical analysis has been applied. In contrary, in Figure 5. it is properly described. In Materials and Methods under 4.15, the statistical analyses for the different evaluations are also not specified.

Lines 152-155: “mdx mice retained more cells in their skeletal muscle than control mice (Table S1). These results suggest that IV-administered WJ-MSCs specifically migrate to the muscles of mdx mice. In contrast, mdx mice had fewer overall residual cells than control mice because of inflammation, which produced a more inhospitable environment in these mice.” This is somewhat contradictory and needs further explanation – why do damaged muscle fibres take up more WJ-MSCs than other tissues as inflammation is predominantly present in the skeletal muscle?

Lines 161-168: “IHC was used to detect collagen I deposition in the myotubes, and this was also significantly increased in the mimic-treated group. However, transfection of the miR-499-5p mimic attenuated myotube fibrosis, with significantly decreased collagen deposition in the miR-499-5p group compared to the untransfected group”. It needs further explanation, what is the mimic treated group?

Line 192: Behavioural tests are not included in this article; therefore, it cannot be stated that behaviour recovery was also present due to the WJ- MSCs administration. In addition, which test was considered to detect levels of apoptosis?

Lines 224-225: “In addition to new advances in targeting TGFβ, targeting fibrosis may prove an important strategy..” “In addition” is not appropriate as TGFβ and fibrosis are in close correlation, actually, TGFβ is a central mediator of fibrosis that acts through TGFβRs.

Line 239: “The mechanism by which human WJ-MSCs regulate miR-499 remains unknown…” Even though if still not well known but possible ways of the therapeutic effect of WJ-MSCs should be explored here. Moreover, other pathways than upregulation of miR-499-5p should be considered and discussed (see (Leibacher and Henschler, 2016, etc.).

General remark:

Authors may consider extending this treatment to other animal models of DMD such as humanized hDMD mice or dystrophin-deficient dogs (cDMD) if available. It is well-known that despite being deficient for dystrophin, mdx mice have minimal clinical symptoms and their lifespan is only reduced by ~25% in contrast to ~75% in humans. Robust skeletal muscle regeneration explains the slowly progressive phenotype of mdx mice. The only exception is the diaphragm, which shows progressive deterioration as is also seen in affected humans. Severe dystrophic phenotypes, such as muscle wasting, etc. do not occur until mice are 15 months or older. Therefore, if no other DMD animal model is available it is suggested to repeat these experiments on older than 2-3 months old mice and on the diaphragm as well.